# MDGCL: Debiased Graph Contrastive Learning with Knowledge of Model Discrepancy

## Abstract

Graph contrastive learning (GCL) have shown promising results for self-supervised representation learning on graph-structured data, benefiting various downstream tasks such as node classification and graph classification. Despite their outstanding performance, a prevalent issue in most existing GCL methods is the arbitrary selection of other data points as negative samples, even when they share the same ground truth label with the anchor. The inclusion of such false negative samples could degrade the performance of GCL. In this study, we present a dual-branch ensembling learning framework, which provides model discrepancy as a crucial indicator to more effectively differentiate false negatives from true negatives. Building on this, we develop a debiased contrastive learning objective. This objective focuses on pulling false negatives closer to the anchor in the embedding space, while simultaneously retaining the capacity to repel true negatives away from the anchor. Extensive experiments on real-world datasets demonstrate the effectiveness of our framework.

## 1 Introduction

Graph neural networks (GNNs) (Kipf & Welling, 2016a; Veličković et al., 2017) have significantly advanced graph representation learning, facilitating various tasks such as node classification, and graph classification. However, their reliance on supervised labels may limit their generalization capabilities (Rong et al., 2019). To address this limitation and achieve more generalizable and transferable representations, self-supervised learning (SSL) has emerged in the field of GNNs. SSL enables GNNs to learn from unlabeled graph data (You et al., 2020; Jin et al., 2020) by training on pretext tasks. Among various SSL techniques, graph contrastive learning (GCL) has gained significant attention due to its impressive empirical performance (Veličković et al., 2019; Hassani & Khasahmadi, 2020b; You et al., 2021; Suresh et al., 2021; Zhu et al., 2020; Thakoor et al., 2021b). Most existing GCL methods adopt an augmentation strategy, which treats augmented versions of the same data as positive samples, and other instances in the same batch as negative samples. Various contrastive objects are studied in the context of graphs, such as node-node (Zhu et al., 2020; Peng et al., 2020), node-(sub)graph (Veličković et al., 2019; Hassani & Khasahmadi, 2020b), and graph-graph (Bielak et al., 2022; Thakoor et al., 2021b; Suresh et al., 2021) contrastive pairs. GCL then aims to maximize the representation similarity between positive pairs and minimize representation similarity between negative pairs.

Despite their great performance, existing GCLs are at risk of noisy views. Due to the absence of labels, most existing GCL methods randomly nodes as negative samples, which raises the risk of introducing noisy views, a situation known as sampling bias (Chuang et al., 2020). The illustration in Figure 3 depicts the strategy employed by GRACE (Zhu et al., 2021). For a given anchor node $v_i$, it designates other nodes as negative samples, which could inadvertently treat nodes of the same class as $v_i$ as negative pairs. The prevalent presence of false negatives significantly hampers the performance of augmentation-based GCL (Xia et al., 2021). Therefore, it is critical to design a denoise framework that can effectively address the widespread issue of sampling bias.

Developing such a framework without label information is challenging. Several initial efforts (Zhao et al., 2021; Zhang et al., 2022a; Xuan et al., 2021; Xia et al., 2021) have been taken to alleviate the effects of

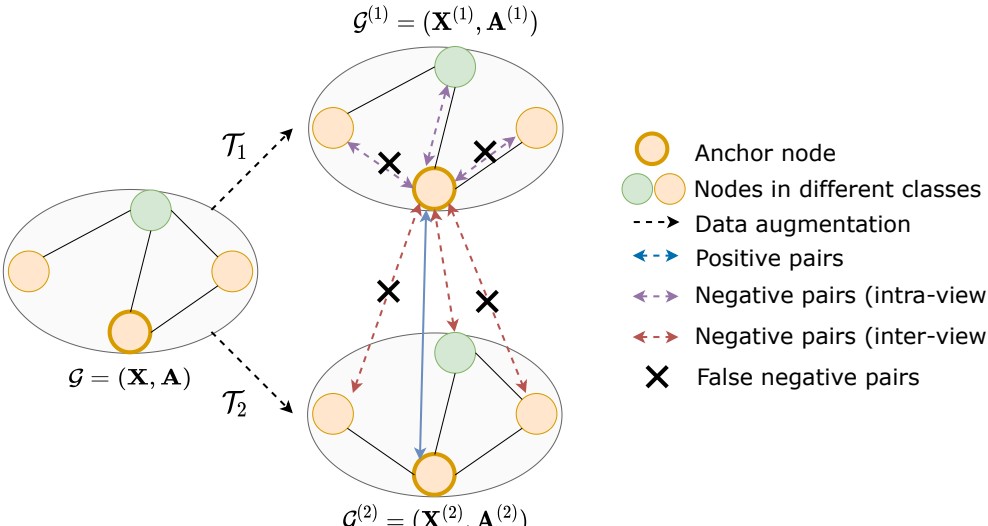

Figure 1: Schematic diagram of node-level GCL framework and illustration of false negative samples in GCL.

false negative views. For example, Zhao et al. (Zhao et al., 2021) adopts clustering results as pseudo labels. However, the quality of pseudo label cannot be guaranteed, which might introduce additional noisy signals. Zhang et al. (Zhang et al., 2022a) generate hard negative samples that are similar to the positive samples but belong to different classes, assuming that by training a model to distinguish between positive samples and hard negatives, the model learns to focus on more subtle and discriminative features. However, the entanglement between hard negative and positive samples can hinder the performance improvement (Xuan et al., 2021). ProGCL (Xia et al., 2021) shows that most negatives with larger similarities to the anchor are false negatives. It fits the distributions of false and true negative samples with Beta Mixture Model (BMM) and assigns weight to each negative sample based on the probability that a negative sample is false. However, for the area where there exists a significant overlap between true negative and false negative samples, it may encounter challenges in determining the appropriate probability that a given sample belongs to each distribution.

As significant overlaps between false negatives and true negatives exist in certain regions, we contend that depending solely on the similarity to the anchor to distinguish a negative sample as false is inadequate. Additionally, current research often overlooks the fact that even with relatively accurate estimations, assigning lower weights to false negatives remains an imperfect solution. In pursuit of high-quality representations for downstream tasks, our goal should focus on bringing false negative samples closer to the anchor rather than pushing them away. However, the work on these is rather limited.

Fortunately, our empirical results in Figure 2 and Figure 3 show that (1) two models trained on different augmented graphs exhibit substantial discrepancy in the similarity of negative samples to the anchor in regions where false and true negative samples significantly overlap; and (2) these two models display a more pronounced discrepancy on true negative samples compared to false negative samples. The details of the preliminary experimental analysis are given in Section III-B. Based on these findings, it appears promising to utilize model discrepancy as a significant indicator to improve our capacity to differentiate between false negatives and true negatives in overlapping areas. Additionally, it paves the way for the formulation of a debiased contrastive objective function which can draw false negatives with the same ground-truth label closer to the anchor.

Therefore, in this paper, we explore a novel problem of estimating the likelihood that a negative sample is false, considering factors beyond mere similarity to the anchor, and draw those false negative samples closer to the anchor. Specifically, we confront two primary challenges: (i) how to integrate additional factors with the commonly used similarity metric to accurately estimate the distributions of unlabeled negative samples; and (ii) how to develop a learning strategy that effectively draws false negative samples closer to the anchor

while retaining the capacity to push true negatives away from the anchor. To tackle these challenges, we introduce a novel framework MDGCL that trains two GNN encoders on distinct augmented graphs. This framework estimates the distribution of negative samples by jointly considering model discrepancy and similarity. With this relatively precise estimation, we selectively sample false negative samples and devise a debiased contrastive learning objective function to maximize the similarity of these samples to the anchor while minimizing the similarity of true negative sample to the anchor. Our main contributions are:

- We propose a novel framework that can better differentiate between false negatives and true negatives by incorporating insights from model discrepancies.

- We design a new debiased contrastive learning strategy aimed at mitigating sampling bias by drawing false negative samples closer to the anchor.

- Extensive experiments on real-world datasets demonstrate the effectiveness of the proposed framework.

## 2 Related work

### 2.1 Graph Contrastive Learning

Graph contrastive learning (Veličković et al., 2019; Hassani & Khasahmadi, 2020b; You et al., 2021; Suresh et al., 2021; Zhu et al., 2020; Thakoor et al., 2021b) has shown great performance for self-supervised representation learning on graphs. Generally, augmentation-based GCL first generate two views of a data sample and prepare positive and negative pairs of each anchor node. It then aims to learn node representations by pushing positive pairs together and pull negative pairs far away. Various methods, including but not limited to DGI (Veličković et al., 2019), HDI (Jing et al., 2021), GMI (Peng et al., 2020), and InfoGCL (Xu et al., 2021), employ this principle by directly quantifying the mutual information (MI) shared across different views. MVGRL (Hassani & Khasahmadi, 2020a) takes this a step further by maximizing the information shared between the cross-view representations of nodes and graphs. Several methodologies such as GRACE (Zhu et al., 2020), GCA (Zhu et al., 2021), ProGCL (Xia et al., 2021), ARIEL (Feng et al., 2022), and gCooL (Li et al., 2022) have successfully implemented the SimCLR framework (Xia et al., 2022) for learning at the node-level. On the graph-level, the SimCLR framework has been effectively utilized by GraphCL (You et al., 2021) and SimGRACE (Xia et al., 2022). Furthermore, innovative CL frameworks, that relieve the dependency on negative samples or data augmentations, have been adopted by BGRL (Thakoor et al., 2021a), AFGRL (Lee et al., 2021), and CCA-SSG (Zhang et al., 2021).

### 2.2 Debiased Contrastive Learning

Despite the great performance of GCL methods, most of them suffer from the problem of sampling bias (i.e. randomly assign other samples as negative samples). Therefore, several efforts (Xia et al., 2021; Liu et al., 2022; Zhao et al., 2021; Chu et al., 2021; Li et al., 2023; Zhu et al., 2022) have been taken to alleviate false negative sample issue. For example, DGCL (Zhao et al., 2021) incorporates clustering pseudo labels to address the issue of false negatives. CuCo (Chu et al., 2021) organizes the negatives from least to most difficult based on similarity in graph-level contrastive learning and introduces a system to automatically select and train negative samples through a curriculum learning framework. ProGCL (Xia et al., 2021) extends GRACE (Zhu et al., 2020) by leveraging hard negative samples via Expectation Maximization to fit the observed node-level similarity distribution. However, the significant overlap between false negatives and true negatives impedes the potential for performance improvement. SpCo (Liu et al., 2022) amplifies the high-frequency components of the augmented graph while retaining its inherent low-frequency structure. HomoGCL (Li et al., 2023) enriches the positive set by incorporating neighboring nodes based on homophily assumption.

Our work is inherently different from existing work: (i) We address the novel challenge of effectively distinguishing false negatives from true negatives in GCL; and (ii) We aim to bring false negative samples with the same ground-truth label closer to the anchor in the embedding space, a critical aspect overlooked by previous works.

## 3 Preliminary

### 3.1 Notation and Background

We use $\mathcal{G} = (\mathcal{V}, \mathcal{E}, \mathbf{X})$ to denote an attributed graph, where $\mathcal{V} = \{v_1, \ldots, v_N\}$ denotes the set of $N$ nodes, $\mathcal{E} \subseteq \mathcal{V} \times \mathcal{V}$ is the set of edges, and $\mathbf{X} = \{\mathbf{x}_1, \ldots, \mathbf{x}_N\}$ is the set of node attributes with $\mathbf{x}_i$ being the node attribute of node $v_i$. $\mathbf{A} \in \mathbb{R}^{N \times N}$ is the adjacency matrix of the graph $\mathcal{G}$, where $A_{ij} = 1$ if nodes $v_i$ and $v_j$ are connected; otherwise $A_{ij} = 0$. Thus, $\mathcal{G}$ can also be denoted as $\mathcal{G} = (\mathbf{X}, \mathbf{A})$. We use $\mathcal{T}$ to denote a random augmentation function, such as randomly dropping edges and masking features.

GCL has become a popular approach for self-supervised representation learning. Generally, it follows the "augmenting-contrasting" learning pattern, where the similarity between two augmentations of the same sample (positive pair) is maximized, while the similarities between two augmentations of different samples (negative pairs) are minimized. The learned node embeddings can be applied to downstream tasks like node classification and node clustering.

Take the popular GCL method GRACE (Zhu et al., 2020) as an example, two augmentation functions $\mathcal{T}_1 \sim \mathcal{T}$ and $\mathcal{T}_2 \sim \mathcal{T}$ are firstly applied to the graph $\mathcal{G}$ to generate two graph views $\mathcal{G}^{(1)} = (\mathbf{X}^{(1)}, \mathbf{A}^{(1)}) = \mathcal{T}_1(\mathbf{X}, \mathbf{A})$ and $\mathcal{G}^{(2)} = (\mathbf{X}^{(2)}, \mathbf{A}^{(2)}) = \mathcal{T}_2(\mathbf{X}, \mathbf{A})$. GRACE then applies a GNN encoder $f_\theta$ to get node embedding for each node in both views as

$$
\begin{aligned}
\mathbf{H}_f^{(1)} &= [f_1^{(1)}, \ldots, f_N^{(1)}] = f(\mathcal{G}^{(1)}), \\
\mathbf{H}_f^{(2)} &= [f_1^{(2)}, \ldots, f_N^{(2)}] = f(\mathcal{G}^{(2)}).
\end{aligned}
\tag{1}
$$

For a node $v_i$, its embedding in one view $f_i^{(1)}$ is regarded as the anchor. The embedding $f_i^{(2)}$ in the other view is the positive sample and the embeddings of other nodes in both views are negatives. GRACE aims to maximize the Mutual Information (MI) between learned representations of positive pairs while minimizing the MI between learned representations of negative pairs by optimizing the following loss function for each anchor as:

$$
\mathcal{L}(v_i, f) = -\log \frac{e^{S\left(f_i^{(1)}, f_i^{(2)}\right)/\tau}}{\underbrace{e^{S\left(f_i^{(1)}, f_i^{(2)}\right)/\tau}}_{\text{positive pair}} + \underbrace{\sum_{k \neq i} e^{S\left(f_i^{(1)}, f_k^{(1)}\right)/\tau}}_{\text{intra−view negative pairs}} + \underbrace{\sum_{k \neq i} e^{S\left(f_i^{(1)}, f_k^{(2)}\right)/\tau}}_{\text{inter−view negative pairs}}},
\tag{2}
$$

where $S(\cdot, \cdot)$ is the cosine similarity and $\tau$ is a temperature parameter.

### 3.2 Preliminary Analysis of GCL Model Discrepancy

To effectively differentiate false negatives, we conduct experiments focusing on the model discrepancy in the representation similarity between false and true negatives relative to anchor points. These experiments lead us to identify model discrepancy as a valuable indicator, paving us a way for the development of a debiased graph contrastive learning framework.

Specifically, we firstly pretrain two 2-layer GNN encoders using GRACE (Zhu et al., 2020) and obtain $f$ and $g$ parameterized by $\Theta_1$ and $\Theta_2$. Given $\mathcal{G}$, we obtain two augmented graphs $\mathcal{G}^{(1)} = \mathcal{T}_1(X, A)$, $\mathcal{G}^{(2)} = \mathcal{T}_2(X, A)$. Then we apply $f$ on $\mathcal{G}^{(1)}$ and $\mathcal{G}^{(2)}$ to inference node representations $\mathbf{H}_f^{(1)}$ and $\mathbf{H}_f^{(2)}$. Similarly, another pretrained GNN encoder $g$ is used to inference node representations $\mathbf{H}_g^{(2)}$ on $\mathcal{G}^{(2)}$.

To measure the difference in similarity between the identical pairs of anchor and its negative sample across $\mathcal{G}^{(1)}$ and $\mathcal{G}^{(2)}$ obtained by network $f$, given an anchor node $v_i$ and its negative sample $v_k$, we calculate the discrepancy as:

$$
abs(S(f_i^{(1)}, f_k^{(1)}) - S(f_i^{(2)}, f_k^{(2)})),
\tag{3}
$$

The discrepancy distribution of $f$ for False Negative and True Negative result are shown in Figure 2 (a).

To measure the difference in similarity between the anchor $v_i$ and its negative sample $v_k$ across encoder $f$ and $g$, we measure the discrepancy as

$$
abs(S(f_i^{(1)}, f_k^{(1)}) - S(g_i^{(2)}, g_k^{(2)})),
\tag{4}
$$

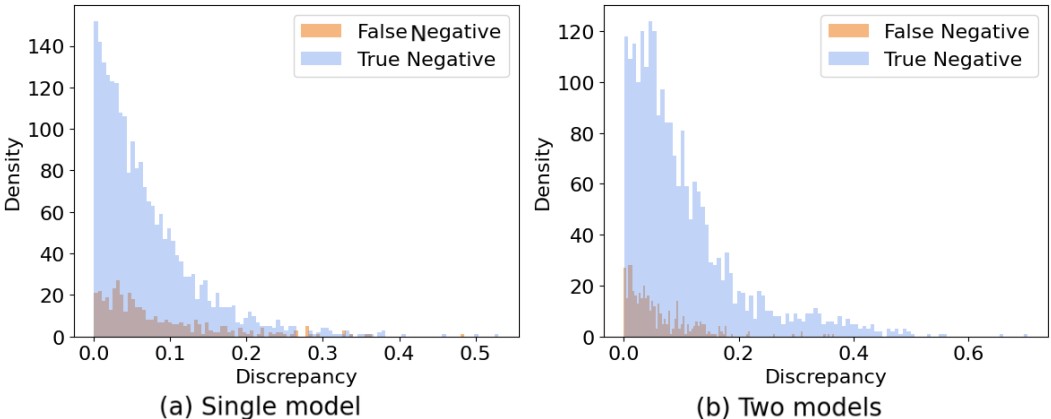

Figure 2: Histograms of discrepancy on similarity (cosine similarity between normalized embeddings of anchor and negatives) on Cora dataset. Given two augmented views $\mathcal{G}^{(1)}, \mathcal{G}^{(2)}$ of the same graph, and GNN networks $f, g$. The discrepancy in (a) is calculated by measuring the difference in similarity between the identical pairs of anchor and its negative sample across $\mathcal{G}^{(1)}$ and $\mathcal{G}^{(2)}$. The discrepancy in (b) is calculated by measuring the difference in similarity between the anchor and its negative sample across networks $f$ and $g$.

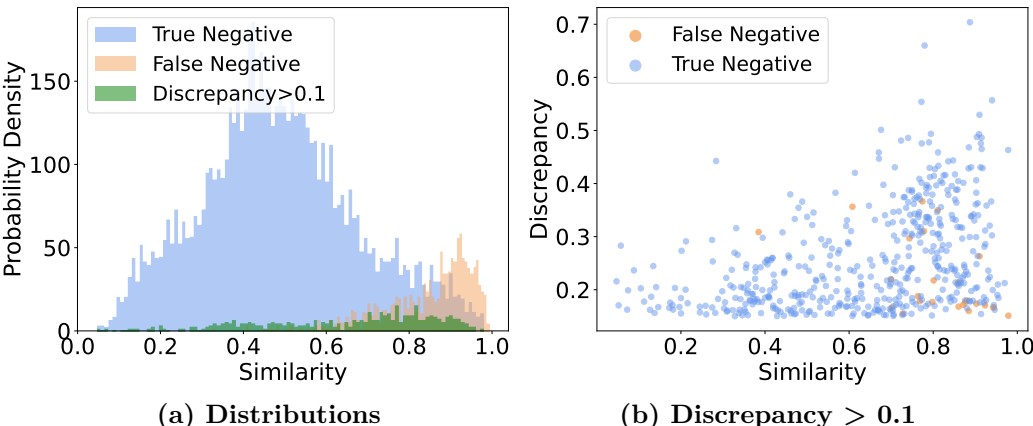

Figure 3: (a) Similarity distributions of false negatives, true negatives and those samples with discrepancy $> 0.1$ on Cora dataset. (b) The distribution of samples with substantial divergence and their corresponding divergence values.

The discrepancy distribution across $f$ and $g$ for False Negative and True Negative is given in Figure 2 (b).

As we observed, the two networks $f$ and $g$ exhibit a more pronounced difference in similarity concerning true negatives compared to false negatives in Figure 2 (b). On the contrary, similar distributional trends in Figure 2 (a) do not provide us with a valuable indicator to distinguish between false and true negatives. *The results indicate that the discrepancy between the two networks, $f$ and $g$, arises not mainly from data augmentation, but from their distinct perceptions of the inherent structure of the data.*

For Figure 3, we further visualize the similarity distributions of false negatives and true negatives considering both $S(f_i^{(1)}, f_k^{(1)}))$ and $S(g_i^{(2)}, g_k^{(2)}))$. Meanwhile, we draw distributions of those samples with discrepancy $> 0.1$ calculated by Equation 4. As shown in (a), while the overall percentage of samples in the overlap region is relatively low, it is noteworthy that the majority of samples displaying large divergence are concentrated within this particular region. Figure 3 (b) illustrates the distribution of samples with substantial divergence

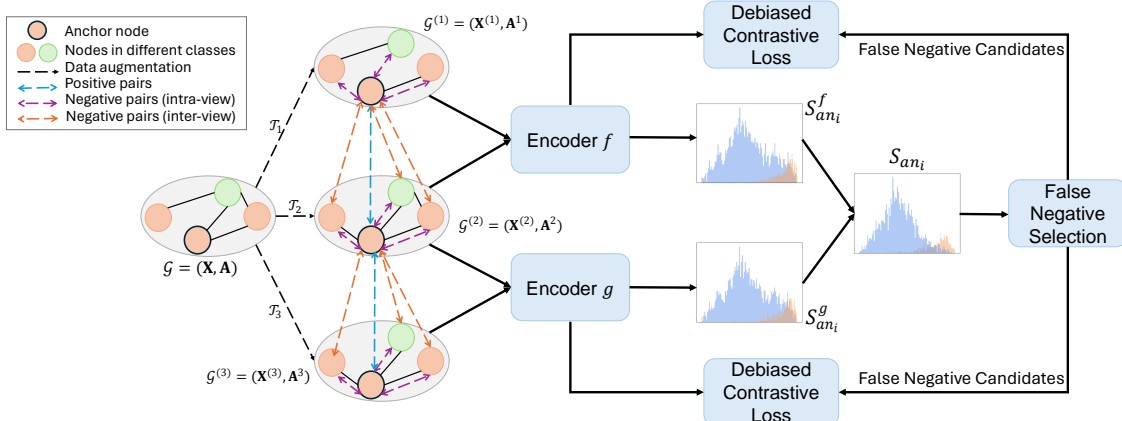

Figure 4: Framework of MDGCL.

and their corresponding divergence values. *It is evident that in the overlapping area, true negatives typically display greater divergence than false negatives.*

This phenomenon can be readily understood as contrastive learning aims to distance negative samples from the anchor in the embedding space. Yet, this becomes complex when hard negative and positive samples are entangled, creating a conflicting dynamic (Xuan et al., 2021). This is due to the exclusive reliance on model parameter adjustments for learning representations. Adjusting parameters to bring positive samples nearer to the anchor results in the inadvertent proximity of entangled hard negatives, which leads to a convoluted dynamic in regions where hard negatives and positives significantly overlap, complicating the update of representations in these areas. Contrarily, false negatives, which share similar semantic information and ground truth label with the anchor, encounter a less intricate learning process. As a result, in overlapping area, it is logical to expect a noticeable divergence between two models that haven been trained on distinct augmented graphs. This divergence arises because random augmentation provides them with the opportunity to learn the inherent structure of data from different perspectives.

The above observations and analyses motivate us to investigate the incorporation of model discrepancy as a vital indicator for distinguishing between false and true negatives, especially in overlap areas where traditional methods struggle to differentiate them effectively. Therefore, in this paper, we study a novel problem of leveraging model discrepancy to counteract sampling bias in graph contrastive learning. The problem is formally defined as:

**Problem 1.** *Given an unlabeled graph $\mathcal{G}$, our goal is to develop a debiased graph contrastive learning framework that incorporates the knowledge of model discrepancy, ensuring that the representations learned through this framework are highly effective in downstream node classification tasks.*

## 4 Methodology

In this section, we present the details of the proposed framework MDGCL, which involves false negative samples selection and a debiased learning strategy to push false negative sample closer to the anchor. An illustration of the proposed framework is given in Figure 4. Building upon our empirical observations that two networks tend to exhibit a greater discrepancy in the overlap area for true negatives compared to false negatives, we adopt a strategy where we train two networks with different augmented views. We then merge the knowledge derived from the model discrepancy along with similarity to the anchor to estimate more distinguishable distributions for false and true negatives. In order to fully harness the distributional information, we employ a sampling approach for false negatives and introduce a novel debiased objective function. This function is designed to maximize the similarity of false negative samples to the anchor, all the while maintaining the original objective of pushing true negative samples away from the anchor within the embedding space. Next, we give the details of each component.

### 4.1 False Negative Selection

To utilize model discrepancy to estimate more discernible distributions for false and true negatives, we adopt two GNN encoders $f$ and $g$ parameterized by $\Theta_1$ and $\Theta_2$, respectively. We train them to learn representations of input view following GRACE (Zhu et al., 2020) separately with loss function in Equation 2. Specifically, given $\mathcal{G}$, during each epoch, we first generate three graph views as

$$\mathcal{G}^{(1)} = T_1(X, A), \ \mathcal{G}^{(2)} = T_2(X, A), \ \mathcal{G}^{(3)} = T_3(X, A) \tag{5}$$

We then apply $f$ on $\mathcal{G}^{(1)}$ and $\mathcal{G}^{(2)}$ to learn node representation as $\mathbf{H}_f^{(1)}, \mathbf{H}_f^{(2)}$. Similarly, we apply $g$ on $\mathcal{G}^{(2)}$ and $\mathcal{G}^{(3)}$ to learn node representation $\mathbf{H}_g^{(2)}, \mathbf{H}_g^{(3)}$. The purpose of having two networks share a common view while also maintaining their own views during training is to enable them to learn from different angles while staying interconnected. This approach guarantees that the observed discrepancies are not merely a product of random data augmentation but primarily arise from the intricate and conflicting overlap between false negatives and true negatives, thereby enhancing the reliability of the discrepancy values.

To streamline our explanation, we will present our framework with node $v_i$, its embedding $f_i^{(1)} \in \mathbf{H}_f^{(1)}$ and $g_i^{(2)} \in \mathbf{H}_g^{(2)}$ as anchor for $f$ and $g$ respectively. Similarly, for node $v_k$, its embedding $f_k^{(1)} \in \mathbf{H}_f^{(1)}$ and $g_k^{(2)} \in \mathbf{H}_g^{(2)}$ as intra-view negative for $f_i^{(1)}$ and $g_i^{(2)}$ respectively. It is worth noting that the same learning strategy is fully applicable to inter-view case as well (i.e. negative samples from $f_k^{(2)} \in \mathbf{H}_f^{(2)}$ and $g_k^{(3)} \in \mathbf{H}_g^{(3)}$ for $f_i^{(1)}$ and $g_i^{(2)}$ respectively).

With learned representations, we combine the similarity as:

$$s_{ik} = \frac{1}{2}(s_{ik}^f + s_{ik}^g) \cdot (1 - d_{ik}), \tag{6}$$

where $s_{ik}^f = S(f_i^{(1)}, f_k^{(1)}), s_{ik}^g = S(g_i^{(2)}, g_k^{(2)})), d_{ik} = abs(s_{ik}^f - s_{ik}^g)$. Going forward this paper, assume that all embedding similarities are Min-Max normalized unless stated otherwise. Then $s_{ik}, d_{ik} \in [0, 1]$. We employ Equation 6 to merge the knowledge of similarity and discrepancy based on the empirical findings previously mentioned. These findings highlight that, generally, the two models exhibit a more substantial divergence on true negative samples compared to false negative samples, especially in overlapping area. As a result, while shifts may occur in both distributions, the shift is more pronounced for true negative samples. This leads to a reduced overlap between the distributions, enhancing the distinction between them. Experiment results in Section 5.4 also verify our analysis.

Different from ProGCL (Xia et al., 2021), which employs the Beta Mixture Model (BMM) to fit the empirical distribution of negative samples, leading to a high computation cost, we utilize $s_{ik}$ as the input for a Bernoulli distribution, from which we sample to determine the selection of a given sample as a false negative. This process is formally defined as:

$$F = \{\mathbf{1}(Bernoulli(s_{ik}) = 1) \mid s_{ik} > \theta, k \in [1, N] \setminus \{i\}\}, \tag{7}$$

where $\theta$ is a predefined threshold, $\mathbf{1}(\cdot)$ is an indication that equals to 1 if the condition holds and 0 otherwise. In this setup, a sample outcome of 1 indicates its selection as a false negative, while a 0 implies it is not selected. Since false negatives usually show smaller differences and higher similarity, while true negatives often display larger discrepancies and lower similarity, our sampling strategy tends to assign more weight to false negatives. Notably, $F$ is shared by $f$ and $g$. By doing so, each time we sample data, there is a significantly higher chance of selecting false negatives, enhancing the effectiveness of our debiased contrastive objective function in the next section.

### 4.2 Debiased Contrastive Learning

Given the indexes of potential false negative candidates, we propose a debiased contrastive learning objective. This objective aims to maximize the similarity of false negatives to the anchor, all the while ensuring that our primary objective of minimizing the similarity of true negatives to the anchor is maintained.

---

**Algorithm 1** MDGCL (intra-view)

---

**Input:** GNN networks $f, g$, augmentation function $\mathcal{T}$, graph $\mathcal{G}$ and normalized cosine similarity function $S$

1: **for** $epoch = 0, 1, 2, ..., E_{max}$ **do**
2:     Draw three augmentation functions $\mathcal{T}_1 \sim \mathcal{T}, \mathcal{T}_2 \sim \mathcal{T}, \mathcal{T}_3 \sim \mathcal{T}$
3:     $\mathcal{G}^{(1)} = \mathcal{T}_1(\mathcal{G}), \mathcal{G}^{(2)} = \mathcal{T}_2(\mathcal{G}), \mathcal{G}^{(3)} = \mathcal{T}_3(\mathcal{G})$
4:     $\mathbf{H}_f^{(1)} = f(\mathcal{G}^{(1)}), \mathbf{H}_f^{(2)} = f(\mathcal{G}^{(2)}), \mathbf{H}_g^{(2)} = g(\mathcal{G}^{(2)}), \mathbf{H}_g^{(3)} = g(\mathcal{G}^{(3)})$
5:     **for all** $f_i^{(1)}, f_k^{(1)} \in \mathbf{H}_f^{(1)}$ **do**
6:         $s_{ik}^f = S(f_i^{(1)}, f_k^{(1)})$
7:     **end for**
8:     **for all** $g_i^{(2)}, g_k^{(2)} \in \mathbf{H}_g^{(2)}$ **do**
9:         $s_{ik}^g = S(g_i^{(2)}, g_k^{(2)})$
10:    **end for**
11:    Compute $s_{ik}$ following Equation 6
12:    Record indexes $F$ of false negative candidates with Equation 7
13:    Compute $\ell(v_i, f), \ell(v_i, g)$ for each anchor $v_i$ for $f$ and $g$ with Equation 9 separately.
14:    Update the parameters of $f$ and $g$ with $\frac{1}{N} \sum \ell(v_i, f)$ and $\frac{1}{N} \sum \ell(v_i, g)$
15: **end for**

**Output:** $f, g$

---

Let $\tau^+$ be the proportion of false negative samples and $\tau^- = 1 - \tau^+$. Given $p$ be the all negative samples distribution and $p_x^+$ be the false negatives distribution. Then the true negatives distribution is calculated as:

$$p_x^- (x) = \left( p(x) - \tau^+ p_x^+(x) \right) / \tau^-. \tag{8}$$

The equation above demonstrates how to mitigate sampling bias by sampling from false negatives. Consequently, when we sample data from the negative distribution $p$ following Equation 8, we inherently draw samples from the true negative distribution $p_x^-$.

Thus, with $F$ representing the indexes for false negative candidates, we sample nodes with corresponding IDs from within this set and denote them as $v_F$. Subsequently, we design our debiased loss function for each anchor as:

$$\ell(v_i, f) = -\log \frac{e^{S_{ii}^f}}{e^{S_{ii}^f} + \frac{N-1}{\tau^-} \underbrace{\left( \frac{1}{N-1} \sum_{i=1}^{N-1} e^{S_{ik}^f} - \tau^+ \frac{1}{M} \sum e^{S_{iF}^f} \right)}_{\text{Debias Sampling}}}, \tag{9}$$

where $S_{ii}^f = S(f_i^{(1)}, f_i^{(2)}), S_{iF}^f = S(f_i^{(1)}, f_F^{(1)})$ and $M = \mid F \mid$ equals to the number of sampled false negatives. We calculate $\tau^+$ as

$$\tau^+ = \frac{M}{N-1}. \tag{10}$$

Here, we aim to highlight the distinctions between our loss function and that in (Chuang et al., 2020). Unlike previous work that sample $v_F$s from positive samples distribution, our approach focuses on sampling from false negatives. This strategic choice is driven by the fact that positive samples typically already exhibit high similarity to the anchor in standard contrastive learning setting. Our objective is not to further enhance the proximity of these positive samples to the anchor – a task relatively easily achieved in standard contrastive learning frameworks. Instead, we aim to specifically target false negatives, endeavoring to draw them closer to the anchor. This approach is intended to effectively mitigate the prevalent issue of sampling bias, addressing a critical gap in existing methodologies.

## 5 Experiments

In this section, we conduct extensive experiments to demonstrate the effectiveness of our proposed MDGCL. In particular, we aim to answer the following research questions:

Table 1: Statistics of datasets used in the paper

| Dataset | #Nodes | #Edges | #Features | #Classes |
|---------|--------|--------|-----------|----------|
| Cora | 2,708 | 10,556 | 1,433 | 7 |
| citepSeer | 3,327 | 9,228 | 3,703 | 6 |
| PubMed | 19,717 | 88,651 | 500 | 3 |
| Photo | 7,650 | 238,163 | 745 | 8 |
| Computer | 13,752 | 491,722 | 767 | 10 |
| ogbn-arXiv | 169,343 | 1,166,243 | 128 | 40 |

- **RQ1** How does our proposed framework perform in terms of node classification accuracy in downstream task?

- **RQ2** Can our framework generate a more distinguishable distribution for false negative and true negative samples?

- **RQ3** Can our framework effectively select false negative samples for debiased contrastive learning?

### 5.1 Experimental Settings

**Baselines.** We compare our method with a variety of representative and state-of-the-art baselines: including supervised graph learning methods GCN (Kipf & Welling, 2016a), GAT (Veličković et al., 2017), graph autoencoders GAE and VGAE (Kipf & Welling, 2016b), augmentation-based GCL methods including DGI (Veličković et al., 2019), MVGRL (Hassani & Khasahmadi, 2020a), CCA-SSG (Zhang et al., 2021), BGRL (Thakoor et al., 2021b), COSTA (Zhang et al., 2022b) , GRACE (Zhu et al., 2020), GCA (Zhu et al., 2021), ProGCL (Xia et al., 2021), ARIEL (Feng et al., 2022), HomoGCL (Li et al., 2023) and SpCo (Liu et al., 2022), augmentation-free graph contrastive learning methods GMI (Peng et al., 2020), AFGRL (Lee et al., 2021) and SUGRL (Mo et al., 2022). The detailed description of the baselines can be found in Appendix A.

**Datasets.** We assess the quality of representations after self-supervised pretraining on six node classification benchmarks, including three citation networks Cora, citepseer, Pubmed (Yang et al., 2016), two co-purchase networks Amazon Computer and Amazon Photo (McAuley et al., 2015) and one large-scale network ogbn-arXiv (Hu et al., 2021). We adopt the public splits for Cora, citepseer and Pubmed, and a 1:1:8 training/validation/testing random splits for the two co-purchase datasets. The statistics of the datasets are provided in Table 1. We give the detailed descriptions in Appendix B.

**Evaluation Protocol.** We follow the linear evaluation scheme as introduced in (Veličković et al., 2019): **i)** We first train the model on all the nodes in a graph without supervision, by optimizing the objective in Equation 9. **ii)** After that, we freeze the parameters of the encoder and obtain node embeddings, which are subsequently fed into a linear classifier (i.e., a logistic regression model) to generate a predicted label for each node. In the second stage, only nodes in training set are used for training the classifier, and we report the classification accuracy on testing nodes. The graph encoder $f$ and $g$ are standard two-layer GCN model (Kipf & Welling, 2016a) for all the datasets. For all experiments, the threshold to filter out false negative samples $\theta$ is as 0.9. For a fair comparison, the hyperparameters of all methods are tuned on the validation set.

### 5.2 Performance Comparison

To answer **RQ1**, we compare node classification performance of our framework with the baselines on various datasets. Each experiment is conducted 10 times. The averaged node classification results with standard deviation are report in Table 2. From the table, we have the following observations:

- InfoNCE-based methods, i.e., GCA (Zhu et al., 2021), ProGCL (Xia et al., 2021), SpCo (Liu et al., 2022) and ARIEL (Feng et al., 2022), cannot bring consistent and significant improvements over GRACE (Zhu et al., 2020). Notably, HomoGCL (Li et al., 2023) consistently yields significant improvements over GRACE. However, our framework surpasses even HomoGCL in performance, which can be attributed

Table 2: Node classification results (accuracy(%)±std). The best and runner-up are marked with boldface and underline, respectively.

| Methods | Cora | CiteSeer | PubMed | Photo | Computer |
|---|---|---|---|---|---|
| GCN | $81.5 \pm 0.4$ | $70.2 \pm 0.4$ | $79.0 \pm 0.2$ | $92.42 \pm 0.22$ | $86.51 \pm 0.54$ |
| GAT | $83.0 \pm 0.7$ | $72.5 \pm 0.7$ | $79.0 \pm 0.3$ | $92.56 \pm 0.35$ | $86.93 \pm 0.29$ |
| GAE | $71.5 \pm 0.4$ | $65.8 \pm 0.4$ | $72.1 \pm 0.5$ | $91.62 \pm 0.13$ | $85.27 \pm 0.19$ |
| VGAE | $73.0 \pm 0.3$ | $68.3 \pm 0.4$ | $75.8 \pm 0.2$ | $92.20 \pm 0.11$ | $86.37 \pm 0.21$ |
| DGI | $82.3 \pm 0.6$ | $71.8 \pm 0.7$ | $76.8 \pm 0.6$ | $91.61 \pm 0.22$ | $83.95 \pm 0.47$ |
| BGRL | $82.7 \pm 0.6$ | $71.1 \pm 0.8$ | $79.6 \pm 0.5$ | $92.80 \pm 0.08$ | $88.23 \pm 0.11$ |
| GMI | $82.4 \pm 0.6$ | $71.7 \pm 0.2$ | $79.3 \pm 1.0$ | $90.73 \pm 0.24$ | $84.22 \pm 0.52$ |
| SUGRL | $83.4 \pm 0.5$ | $\underline{73.0 \pm 0.4}$ | $\underline{81.9 \pm 0.3}$ | $\underline{93.07 \pm 0.15}$ | $\underline{88.93 \pm 0.21}$ |
| AFGRL | $79.8 \pm 0.2$ | $69.4 \pm 0.2$ | $80.0 \pm 0.1$ | $92.71 \pm 0.23$ | $88.12 \pm 0.27$ |
| COSTA | $82.2 \pm 0.2$ | $70.7 \pm 0.5$ | $80.4 \pm 0.3$ | $92.43 \pm 0.38$ | $88.37 \pm 0.22$ |
| MVGRL | $82.9 \pm 0.6$ | $72.6 \pm 0.5$ | $79.8 \pm 0.5$ | $91.66 \pm 0.42$ | $87.07 \pm 0.63$ |
| CCA-SSG | $84.0 \pm 0.4$ | $\mathbf{73.1 \pm 0.3}$ | $81.0 \pm 0.4$ | $92.84 \pm 0.18$ | $88.27 \pm 0.32$ |
| GRACE | $81.5 \pm 0.3$ | $70.6 \pm 0.5$ | $80.2 \pm 0.3$ | $92.15 \pm 0.24$ | $86.25 \pm 0.25$ |
| GCA | $81.4 \pm 0.3$ | $70.4 \pm 0.4$ | $80.7 \pm 0.5$ | $92.53 \pm 0.16$ | $87.80 \pm 0.23$ |
| ProGCL | $81.2 \pm 0.4$ | $69.8 \pm 0.5$ | $79.2 \pm 0.2$ | $92.39 \pm 0.11$ | $87.43 \pm 0.21$ |
| ARIEL | $83.0 \pm 1.3$ | $71.1 \pm 0.9$ | $74.2 \pm 0.8$ | $91.80 \pm 0.24$ | $87.07 \pm 0.33$ |
| HomoGCL | $\underline{84.3 \pm 0.5}$ | $72.3 \pm 0.7$ | $81.1 \pm 0.3$ | $92.92 \pm 0.18$ | $88.46 \pm 0.20$ |
| SpCo | $82.7 \pm 0.6$ | $71.3 \pm 0.8$ | $81.0 \pm 0.4$ | $92.74 \pm 0.17$ | $88.14 \pm 0.28$ |
| MDGCL | $\mathbf{84.6 \pm 0.5}$ | $\mathbf{73.1 \pm 0.5}$ | $\mathbf{82.3 \pm 0.5}$ | $\mathbf{93.26 \pm 0.28}$ | $\mathbf{89.14 \pm 0.22}$ |

that HomoGCL arbitrarily designates non-neighboring nodes as negative samples, thereby introducing false negative pairs.

- Our superior performance compared to ProGCL (Xia et al., 2021) demonstrates the effectiveness of our framework in generating distinguishable distributions for false and true negatives.

- Among augmentation-free methods such as GMI (Peng et al., 2020), SUGRL (Mo et al., 2022) and AFGRL (Lee et al., 2021), SUGRL stands out with its competitive performance. While it almost outperforms all other augmentation-based frameworks, it falls short only to our MDGCL. The result indicates that data augmentation remains necessary in GCL, but we need to adopt debiased framework to learn better node representations.

## 5.3 Results on Large-Scale OGB Dataset

To show that the scalability of the proposed framework, we also conduct an experiment on a large-scale dataset ogbn-arxiv from OGB benchmark (Hu et al., 2021). Following (Hu et al., 2021), we extend the backbone GNN encoder to 3 GCN layers, we report the classification accuracy on both the validation and test sets, which is a convention for this task. The results are shown in Table 3. The results show our framework outperforms all other unsupervised learning methods, which demonstrates the effectiveness and scalability of the proposed method.

## 5.4 Distributions Comparison

To answer **RQ2**, in this subsection, we show empirical results of how distributions of false and true negative samples shift when incorporating the knowledge of model discrepancy with our strategy. The results are shown in Figure 5. Similar to Section III-B, our experiment initially involves pretraining both $f$ and $g$, followed by the computation of the similarity scores $\frac{1}{2}(s_{ik}^f + s_{ik}^g)$ displayed on the left side of each subfigure. On the right side of each subfigure, we present values obtained by multiplying these similarity scores by $(1 - d_{ik})$, denoted as $\frac{1}{2}(s_{ik}^f + s_{ik}^g) \cdot (1 - d_{ik})$. From the figure, we can observe that our strategy of combining model discrepancy with similarity results in a reduced overlap between false negatives and true negatives. This is advantageous for the task of distinguishing false negatives from true negatives efficiently.

Table 3: Node classification results on ogbn-arXiv dataset (accuracy(%)±std). OOM indicates out-of-memory.

| Model | Validation | Test |
|---|---|---|
| MLP | 57.65±0.12 | 55.50±0.23 |
| GCN | 73.00±0.17 | 71.74±0.29 |
| GraphSAGE | 72.77±0.16 | 71.49±0.27 |
| Random-Init | 69.90±0.11 | 68.94±0.15 |
| DGI | 71.26±0.11 | 70.34±0.16 |
| GRACE full-graph | OOM | OOM |
| GRACE-Subsampling (k=2) | 60.49±3.72 | 60.24±4.06 |
| GRACE-Subsampling (k=8) | 71.30±0.17 | 70.33±0.18 |
| GRACE-Subsampling (k=2048) | 72.61±0.15 | 71.51±0.11 |
| ProGCL | 72.45±0.21 | 72.18±0.09 |
| BGRL | 72.53±0.09 | 71.64±0.12 |
| HomoGCL | 72.85±0.10 | 72.22±0.15 |
| MDGCL | **72.93±0.12** | **72.33±0.13** |

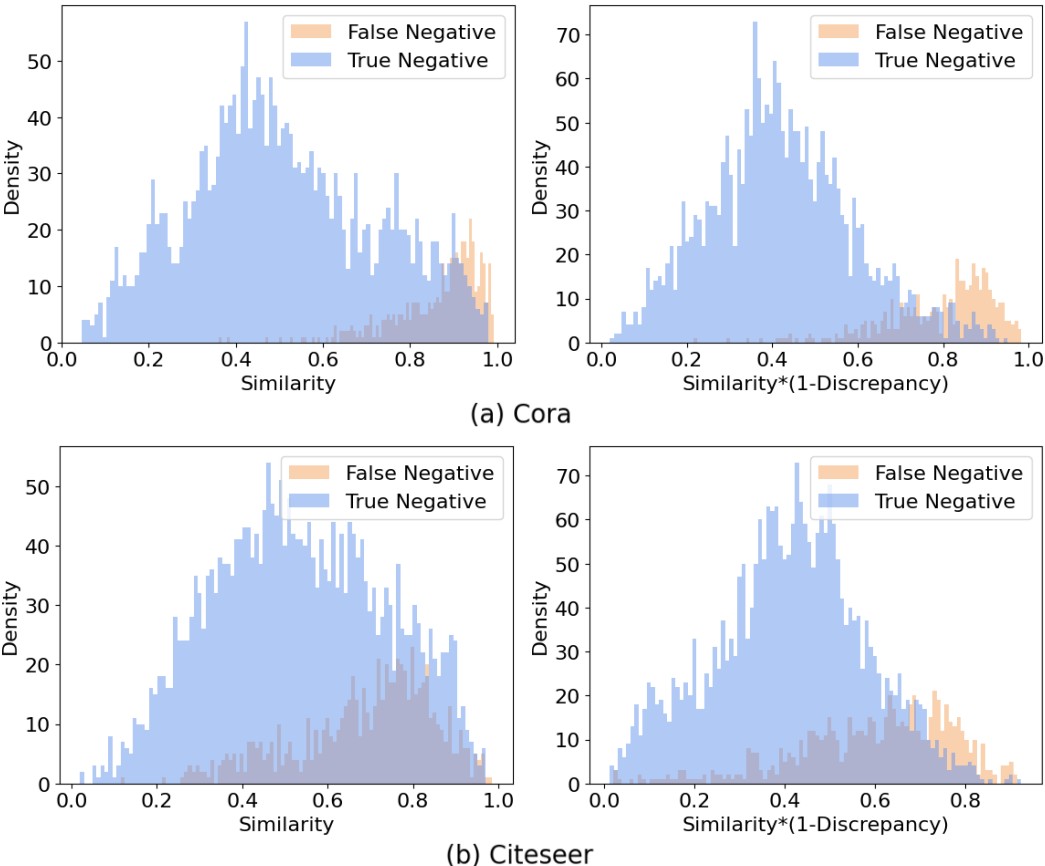

Figure 5: Comparison of distributions using solely similarity and combining the knowledge of similarity and discrepancy on (a) Cora and (b) citepseer.

## 5.5   Ability of Selecting False Negatives

To show that the proposed framework can effectively mitigate sampling bias, in this subsection, we conduct experiments to quantify the percentage of false negatives sampled with Equation 7, which answers **RQ3**.

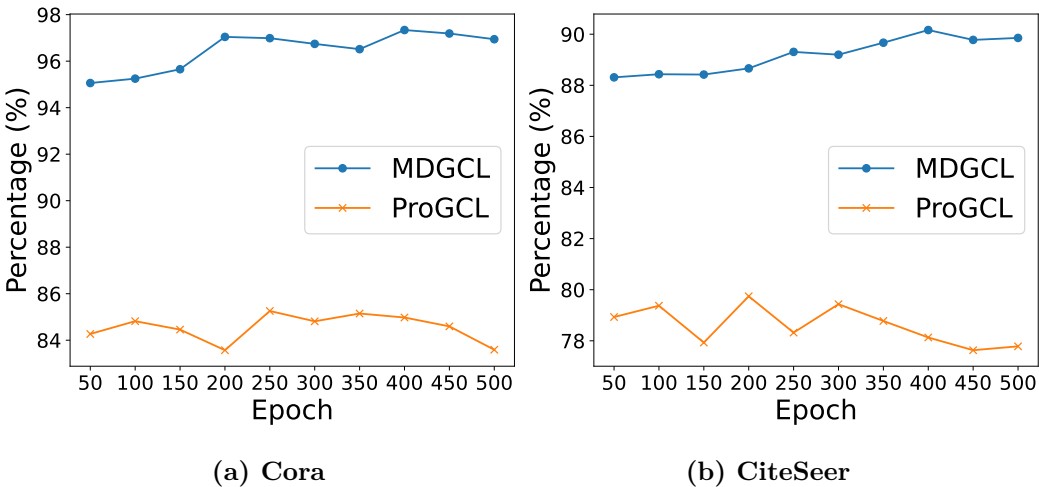

**(a) Cora**            **(b) CiteSeer**

Figure 6: Case study on the percentage of selected false negatives on (a) Cora and (b) CiteSeer.

Specifically, we first train our model then calculate the percentage of false negatives using the ground-truth labels. Note that the labels are not used during model training but are only used when calculating percentage of false negatives. We compare our method with the approach used in ProGCL (Xia et al., 2021), which exclusively relies on similarity as the indicator for distinguishing false negatives from true negatives. The results are shown in Figure 6. From the figure, we can observe that, when combining the measure of discrepancy with similarity, our framework consistently samples a higher percentage of false negatives. This increase in the proportion of false negatives enhances the effectiveness of our debiased objective function in Equation 9, effectively bringing false negative samples with the same ground-truth label closer to the anchor.

## 5.6 Ablation Study

In this section, we conduct ablation study to understand the contribution of each component of MDGCL. We compare the performance of our MDGCL with (i) **MDGCL/M**: MDGCL without model discrepancy as additional knowledge for beta distribution estimation; and (ii) **MDGCL/D**: MDGCL without debiased objective function but adopt the loss function in Eq. (10) in ProGCL (Xia et al., 2021). We only show the results on Cora and Pubmed as similar trends are observed on other datasets. The results are presented in Fig. 7. From this figure, we observe that: (i) Consistently, MDGCL outperforms MDGCL/D, underscoring the effectiveness of the strategy of bringing false negative samples closer to the anchor. This approach proves more successful than assigning lower weights to push them away from the anchor; (ii) MDGCL/D consistently outperforms MDGCL/M, highlighting that a more distinguishable distribution forms the foundation of our framework. This distinction is crucial because, without a discernible distribution, a significant portion of true negatives might inadvertently be selected to be drawn closer to the anchor, potentially introducing other noise into the process; and (iii) MDGCL consistently outperforms both MDGCL/M and MDGCL/D, confirming that our approach of generating a discernible distribution and designing a denoising objective function is indeed sound.

## 5.7 Hyperparameter Sensitivity Analysis

In this section, we conduct experiments to show how the negative sample selection threshold $\theta$ impacts the performance of MDGCL. We vary the values of $\theta$ as $\{0.75, 0.8, 0.85, 0.9, 0.95\}$ for both Cora and Pubmed. We run experiments for each parameter five times and report the mean accuracy in Table 4. We have observed the followings: (1) To filter out false negative samples while preserving true negative samples, it is recommended to set $\theta \in [0.85, 0.95]$. (2) When $\theta \geq 0.85$, our MDGCL typically achieves high performance, which eases hyperparameter tuning.

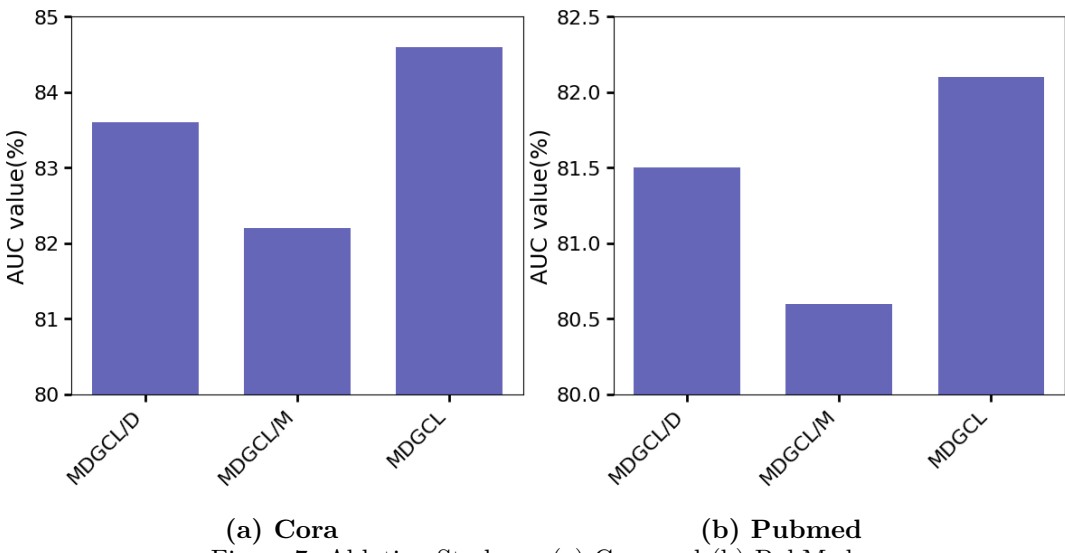

**(a) Cora**           **(b) Pubmed**

Figure 7: Ablation Study on (a) Cora and (b) PubMed.

## 5.8 Time Complexity Analysis

Our framework is easily parallelizable, as the message passing process is independent for each network $f$ and $g$. Consequently, the time complexity of MDGCL can be reduced to be the same as that of GRACE (Zhu et al., 2021). This proves that MDGCL has great potential in conducting scalable graph contrastive learning.

Table 4: Node classification results (accuracy(%)±std) for Hyperparameter Sensitivity Analysis.

|        | 0.75 | 0.80 | 0.85 | 0.90 | 0.95 |
|--------|------|------|------|------|------|
| Cora   | 83.2 | 83.6 | 84.1 | 84.6 | 84.3 |
| Pubmed | 80.7 | 81.2 | 81.7 | 82.3 | 81.7 |

## 6 Conclusion

In this paper, we introduce a novel debiased graph contrastive learning framework aimed at mitigating the prevalent issue of sampling bias. Central to our approach is the incorporation of model discrepancy, which facilitates the generation of a more distinct distribution between false and true negative samples. This enhancement significantly improves our capacity to identify false negatives. Furthermore, we have developed a unique loss objective that effectively draws false negatives towards the anchor while simultaneously repelling true negatives. Through comprehensive experiments, our framework has demonstrated superior performance in downstream node classification tasks, outperforming current state-of-the-art methods in terms of accuracy.

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

## A  Baselines

- GCN (Kipf & Welling, 2016a), GAT (Veličković et al., 2017), GraphSAGE (Hamilton et al., 2018): GCN, GAT, GraphSAGE are three popular supervised GNNs.

- GAE/VGAE (Kipf & Welling, 2016b): GAE and VGAE are graph autoencoders that learn node embeddings via vallina/variational autoencoders. Both the encoder and the decoder are implemented with graph convolutional network.

- DGI (Veličković et al., 2019): It maximizes the mutual information between patch representations and corresponding high-level summaries of graphs obtained from graph convolutional network.

- GMI (Peng et al., 2020): GMI applies cross-layer node contrasting and edge contrasting. It also generalizes the idea of conventional mutual information computations to the graph domain.

- MVGRL (Hassani & Khasahmadi, 2020a): MVGRL maximizes the mutual information be- tween the cross-view representations of nodes and graphs using graph diffusion.

- BGRL (Thakoor et al., 2021a): BGRL adopts asymmetrical structure to do the node-node level contrast without negative samples to avoid quadratic bottleneck.

- AFGRL (Lee et al., 2021): It extends BGRL by generating an alternative view of a graph by discovering nodes that share the local structural information and global semantics with the graph.

- CCA-SSG (Zhang et al., 2021): CCA-SSG leverages classical Canonical Correlation Analysis to construct feature-level objective which can discard augmentation-variant information and prevent degenerated solutions.

- COSTA (Zhang et al., 2022b): COSTA alleviates the highly biased node embedding obtained via graph augmentation by performing feature augmentation.

- GRACE (Zhu et al., 2020): It adopts SimCLR which performs graph augmentation on the input graph and considers node-node level contrast on both inter-view and intra-view levels.

- GCA (Zhu et al., 2021): GCA extends GRACE by considering adaptive graph augmentations based on degree centrality, eigenvector centrality, and PageRank centrality.

- ProGCL (Xia et al., 2021): ProGCL extends GRACE by leveraging hard negative samples via Expectation Maximization to fit the observed node-level similarity distribution.

- ARIEL (Feng et al., 2022): It extends GRACE by introducing an adversarial graph view and an information regularizer to extract informative contrastive samples within a reasonable constraint.

- SUGRL (Mo et al., 2022): SUGRL uses multiplet loss to boost interclass variation by integrating structural and neighbor information. It also adds an upper bound loss to limit the distance between positive and anchor embeddings, thereby reducing intra-class variation.

- SpCo (Liu et al., 2022): It optimizes the contrastive pair with the original adjacency matrix and elevates augmented graph's high frequency while preserving its original low frequency structure.

- HomoGCL (Li et al., 2023): It is a model-agnostic framework that enhances the positive set with significant neighbor nodes.

## B  Datasets

- **Cora**, **CiteSeer**, and **PubMed** (Yang et al., 2016): They are citation networks where nodes denote papers, and edges depict citation relationships. In Cora and CiteSeer, each node is described using a binary word vector, indicating the presence or absence of a corresponding word from a predefined dictionary. In contrast, PubMed employs a TF/IDF weighted word vector for each node. For all three datasets, nodes are categorized based on their respective research areas.

- **Amazon-Photo** and **Amazon-Computers** (McAuley et al., 2015): In these networks, nodes correspond to products, and edges indicate co-purchase instances. Each node is characterized by a raw bag-of-words feature, which encodes product reviews, and is labeled according to its product category.

- **ogbn-arXiv** (Hu et al., 2021): It is a citation network encompassing all Computer Science arXiv papers cataloged in the Microsoft Academic Graph. Each node is characterized by a 128-dimensional feature vector, which is derived by averaging the skipgram word embeddings present in its title and abstract. Additionally, the nodes are categorized based on their respective research areas.

