# OpenReview forum: "MDGCL: Debiased Graph Contrastive Learning with Knowledge of Model Discrepancy"
_TMLR — Rejected by TMLR_

### Review · Reviewer_P4bT · 2024-04-14

**Summary Of Contributions:**

This work studies the sampling biases in graph contrastive learning (GCL). The authors find that the augmentation in GCL may introduce some false negative examples and further affect the optimization and the performance. Therefore, they propose a new method called MDGCL to mitigate the issues of the false negatives by leveraging the knowledge of model discrepancy. They conduct extensive experiments and analysis to demonstrate the effectiveness of MDGCL.

**Audience:**

Yes

**Broader Impact Concerns:**

There is no discussion of the broader impacts.

**Claims And Evidence:**

No

**Requested Changes:**

1. The presentation is not sufficiently clear:
- In fact, there seems not a clear definition for how to find the true/false negatives in the preliminary analysis;
- Some notations like $f$ are used multiple times for different meanings;
- In Table 1, "citepSeer";

2. The experiments are limited to certain datasets:
- The current analysis is limited to the final learned representations, which may not properly reflect the influence of false negative examples on training;
- The analysis and ablation studies are limited to simple datasets, while it's unclear whether the observation still hold for larger graphs, or heterophilous graphs;
- The experiments are limited to homophilous graphs;

3. The discussion with some related works that also study the biases or debiasing GCL[1,2] is missing.

**References**

[1] Calibrating and Improving Graph Contrastive Learning, TMLR'23.

[2] B2-Sampling: Fusing Balanced and Biased Sampling for Graph Contrastive Learning, KDD'23.

**Strengths And Weaknesses:**

(+) The false negative example is a huge concern in GCL;

(+) The authors conduct extensive analysis the demonstrate the idea of MDGCL.

(-) The presentation is not sufficiently clear;

(-) The experiments are limited to certain datasets;

(-) The discussion with some related works are missing;

---

### Review · Reviewer_j8hr · 2024-04-16

**Summary Of Contributions:**

This paper aims to tackle the problem of sampling bias in graph contrastive learning, which stems from the arbitrary selection process of negative samples that may include data points with the same ground truth label as the anchor (called false negatives). To tackle this limitation, the authors propose to use the model discrepancy of different GNNs, which is used as an indicator to differentiate false negatives from true negatives. In addition, the authors propose a new objective, which aims to pull false negatives closer to the anchor in the embedding space while repelling true negatives away from the anchor. The authors experimentally validate the proposed method, namely MDGCL, on node- and graph-classification benchmark datasets, showing its advantage over baselines.

**Audience:**

Yes

**Broader Impact Concerns:**

The authors do not discuss any concerns about the ethical implications of their work; however, I don't see any major concerns about them.

**Claims And Evidence:**

Yes

**Requested Changes:**

Please see the weaknesses above.

**Strengths And Weaknesses:**

### Strengths
* This paper tackles the important and interesting problem of sampling bias (i.e., distinguishing false negatives from true negatives) during graph contrastive learning.
* The proposed method of using model discrepancies (to address the sampling bias problem) is well supported by the analyses in Figures 2 and 3.
* The experimental setups (especially the baselines that the authors compare) are extensive, further including both the node- and graph-classification tasks.
* This paper is well-written and easy to follow.

### Weaknesses
* There is a recent graph self-supervised learning approach that is not impacted by the sampling bias problem (as it leverages the distances between two graphs for learning), namely D-SLA [1], which may be worthwhile to discuss and compare.
* The performance improvements of the proposed MDGCL over baselines are marginal and seem not statistically significant. The authors may additionally perform statistical tests (e.g., t-test with a p-value of 0.05), to showcase the effectiveness of the proposed method.
* In Table 4, it may be a typo that the authors do not provide the standard deviations, which contradicts with its caption.

---

[1] Graph Self-supervised Learning with Accurate Discrepancy Learning, NeurIPS 2022.

---

### Review · Reviewer_g8rD · 2024-04-30

**Summary Of Contributions:**

This paper aims to find false negative samples in GCL. The authors observe that model discrepancy can be used to find these negative samples. They further propose MDGCL to bring false negative samples closer to the anchor.

**Audience:**

Yes

**Broader Impact Concerns:**

No direct limitations and potential negative societal impact

**Claims And Evidence:**

Yes

**Requested Changes:**

Requested Changes

Q1: In Figure 2, 3 and 5, the authors use ‘density’  or ‘probability density’ as the label of y-axis. However, the histograms are not normalized. As observed from these figures, there are more true negative samples compared to false negative samples. Thus it is hard to see the distribution discrepancy in these figures. The authors should also add theoretical analysis or qualitative results for the phenomenon described in Figure 2 and 3.
Q2: Strong baselines are not included in Table 3. For example, CCA-SSG is in Table 2 but not in Table 3. Please compare MDGCL with these methods on the Ogbn-Arxiv dataset.
Q3: Please include runtime comparison on the Ogbn-Arxiv dataset. It seems like MDGCL needs to train GCL 3 times, the runtime could be very long. In Section 5.8, the authors claim “the time complexity of MDGCL can be reduced to be the same as that of GRACE. This proves that MDGCL has great potential in conducting scalable graph contrastive learning. ” The second sentence is not true since GRACE is known to be very time-consuming. Please modify.

Typo: CiteSeer is misspelled to ‘citepSeer’ multiple times. The authors seem to have used replace all in some text editor.

**Strengths And Weaknesses:**

Strengths And Weaknesses

S1: Finding false negative samples is important in GCL.
S2: Use model discrepancy to find negative samples seems novel.
S3: The literature review is detailed.

W1: The visualization in this paper needs improvement.
W2: The motivation, while interesting, is not theoretically supported.
W3: The scalability experiment is inadequate.

---

### Decision · Action_Editor_mzwU · 2024-06-24

**Recommendation:** Reject

**Comment:**

The paper proposes MDGCL to identify false negative samples in GCL using model discrepancy. While this approach appears novel, the overall contribution is considered weak. Empirical evaluations are unsatisfactory, with marginal and statistically insignificant improvements. The paper has poor visualization, lack of theoretical support, and inadequate scalability experiments. It also fails to discuss recent relevant approaches like D-SLA. Plus, the authors failed to provide a rebuttal. Overall, the paper is clearly under the bar of TMLR.

**Audience:**

Yes. Graph learning community would appreciate reading this paper.

**Claims And Evidence:**

The claims in the submission are not fully supported by accurate, convincing, and clear evidence. The overall novelty is weak despite the novel use of model discrepancy. Empirical evaluations are unsatisfactory, with marginal, statistically insignificant improvements. The visualization needs improvement, and the motivation lacks theoretical support. The scalability experiment is inadequate, and there are inconsistencies in the reported results (e.g., missing standard deviations in Table 4). Additionally, the paper fails to compare with recent relevant approaches like D-SLA. These issues collectively undermine the paper's credibility and effectiveness.